# Proposals on Conformal Cyclic Evolution of The Universe

Natarajan Shriethar[1]

[1]Department of Energy Science, Alagappa University, Karaikudi, India , natarajangravity@gmail.com

## Abstract

On the conformal cyclic evolution of the universe, two ideas are proposed. With the help of loop quantum gravity, loop quantum cosmology and conformal cyclic cosmology these postulates are proven to exist. To support the postulates a full universe hypothesis is also proposed. Without facing big rip singularity, a phantom dominated conformal cyclic universe is discussed. The article is written in simplest possible manner for everyone to understand it.

**Keywords** Loop quantum cosmology, Loop quantum gravity, Conformal cyclic cosmology, Dark energy, Phantom energy, Dark matter

## 1   Introduction

To explain the evolution of the universe, there exist many cosmological models. For example universe from vacuum model [14] [34], the mathematical universe model [33], ekpyrotic model [17] and the braneworld cosmological model [15], are exist to explain the evolution of the universe. This work proposes two postulates on the conformal evolutional theory of the universe. The proposed postulates let to define the super universe hypothesis and non-beginning, a non-ending scenario of the universe. To prove and confirm the dark energy interaction in higher scales, some data from IllustrisTNG are implemented. The relationship between dark matter and the gravitational interaction is plotted. These data are discussed elusively in later sections. This article is written in the following way. It begins with the postulates. To understand the further concepts which will support the postulates, some basic understanding of certain topics is necessary. Hence basic introduction on loop quantum gravity and conformal cyclic cosmology are briefed in the sections 3 and 4. The proofs for the postulates discussed in the section 2 is discussed in 5 and 6 respectively. Various results are discussed in section 7. For the birth and end of the universe, the fate of any universe is said to be in a big bang or big rip. The work presents the proof for the proposed postulates from previously published solutions by us.

# 2   Postulates

To understand the conformal cyclic evolution of the universe there are two postulates are proposed. These postulates attempt to provide a glimpse of the solutions for the various problems that exist in the standard model of cosmology.

1. For the super universe big bang may not be the actual beginning.

2. At the big rip a conformal cycle of a universe would not have a necessity to diverge.

These postulates explain the quantum gravitational domination over any possible evolution of the universe. Based on these postulates combined models of the universe may be constructed. To understand the proofs of the postulates, some introduction to loop quantum gravity and conformal evolution of the universe is required.

The necessary theoretical explanations and proofs for these postulates are given in this context. Based on theories such as loop quantum cosmology, conformal cyclic cosmology and loop quantum gravity, a general way of understanding the whole picture is proposed via these postulates. Before reaching the proofs for these postulates some introduction on basic concepts to understand the proofs are given in the next section.

# 3   Loop quantum gravity

The various scale factors, matter density and pressure at the singularity can be resolved with quantum gravitational theory. Especially the loop quantum formalism discusses the evolution of the universe in quantum levels with mathematical consistencies [12] [9]. The singularities can not be understood via classical general relativity. Similarly, the general relativity could not explain the Planckian level and trans-Planckian level behaviour of spacetime. As the quantum effects dominate in such a regime, the classical evolutions break down at the singularity. Hence the quantum version of spacetime theory is needed to explain the quantum nature of gravity.

The quantum gravitational theories attempt to zoom into the singularity and predicts its kinematical evolution at quantum levels. In general, the quantization takes place by promoting classical phase-space into Hilbert space and phase-space variables as quantum mechanical operators. And the equations of motion are promoted as constraint equations. In loop quantum gravity the quantization of spacetime itself is attempted. To do so there are a new set of variables are introduced. Those variables are called as Ashtekar variables [3], [6]. These variables quantize the spacetime with triads and connections, they allow one to understand the quantum kinematical behaviour of spacetime. Similarly, the constraint equations resolve the Hamiltonian in the quantum spacetime level. The loop quantum gravity has great success on the resolution of spacetime singularities. The big bang singularity is replaced with big bounce solutions which are offed by loop quantum cosmology [5], [1].

Similarly the black hole singularity is resolved by loop quantum gravitational solutions [26], [18]. These solutions suggest that quantum evolution extends beyond

the singularity without being perturbed. Like loop quantum evolution resolves the big bang and black hole singularity, every possible singularity in the evolution are been resolved [4]. The future singularities are also resolved by loop quantum cosmological formalism [31]. Every strong singularity is excluded from the evolution of the universe using loop quantum cosmology. Especially the big rip singularity is resolved using loop quantum cosmological paradigm [29]. As the big rip singularity is avoided in the LQG, then the future evolution of the universe beyond the big rip can be predicted. The quantum effects dominate the evolution of the universe. Hence the quantum information within the universe will remain unperturbed, and that will be transformed into future aeons. By fusing the loop quantum cosmology theory with conformal cyclic cosmology, the proposed postulates are discussed in the present work. A brief overview of conformal cyclic cosmology is presented in the next section.

# 4    Conformal cyclic cosmology

In conformal cyclic cosmology, the transformation from the big bang to the end of the evolutional trajectory is considered as a cycle of a universe. These cycles are defined as aeons in conformal cyclic cosmology. With the existing proofs and data, in alternative to the big bang model, the conformal cyclic model is proposed by Roger Penrose [24]. The idea behind the CCC is the gravitization of quantum mechanics as suggested by him [25]. The CCC proposes a conformal mapping between two successive singularities. To understand the conformal evolution in terms of mathematical conformal mapping, the expanded final state of the universe is to be compressed and the initial state of the universe is to be expanded.

These two different stages of the universe are mapped together on a 2-surface that connects both of them. Such fusing allows the universe to evolve conformally without stopping its evolution by facing any singularities. The conformal evolution can be added up with the loop quantum cosmological formalism and they give proof for the postulates suggested in the current work. The CCC has numerous proofs in terms of Hawking points and Cosmic Microwave Background. The existence of a conformal cyclic cosmological model is predicted by some experimental proofs. Among them, the observation of Hawking points distributes the strong holding for the model [2]. It is also believed that the CCC model will be a solid candidate for the standard cosmological theories in future. The CCC attempts to unify the dark matter, black holes and conformal evolution of the universe. The CCC model avoids the necessity of inflation during the expansion phase too.

# 5    Proof for postulate I

postulate I suggests that the big bang might not be the actual beginning of the super universe. From the solutions of Einstein field equations, the big bang singularity is obtained mathematically. With plenty of cosmological solutions, the big bang model for the evolution of the universe is formulated. The model suggested that the universe emerged from the big bang singularity and expand exponentially into higher volumes. As from the big bang model the earlier universe could be in the

hot and denser stages. As the universe expands the density of the universe becomes lesser and the volume grows up to higher magnitudes. The evolution of the universe is predicted by FRW equations.

$$\left(\frac{\dot{a}}{a}\right)^2 = \frac{8\pi G}{3}\rho - \frac{kc^2}{a^2} + \frac{\Lambda c^2}{3} \tag{1}$$

Here $a$ is the scale factor, $\Lambda$ is the cosmological constant, $k$ is the curvature parameter and $\rho$ is the density parameter.

$$\frac{\ddot{a}}{a} = -\frac{4\pi G}{3}\left(\rho + \frac{3p}{c^2}\right) + \frac{\Lambda c^2}{3} \tag{2}$$

Here $p$ is the pressure. There are plenty of observational proofs available for the existence of this cosmological model. Cosmic microwave background data [11] and supernova observations [27] are some proofs for the big bang cosmology. The expansion of the universe and dominant components of the universe are explained from equation 1, 2.

The big bang singularity has quantum mechanical properties. Hence it is resolved with loop quantum gravitational paradigm. The loop quantum gravity resolves the big bang singularity as big bounce solutions. The classical continuum universe is resolved with quantum gravitational discrete solutions. It resolves the wavefunction of the universe and explains with fundamental chunks of spacetime. In such a view if the time-reversal of curvature is done in the view of loop quantum gravity. The massless scalar field $\phi$ which acts as time provides the possible evolution beyond the big bang singularity. Hence the big bang is viewed as a bounce that has a dual solution in both forward and reverses time. From loop quantum solutions themselves, one can obtain the fact that the big bang is not just an actual beginning. The cyclic evolution is understood from this transition point. As the reversible of time is assumed throughout every possible previous cycle of the universe, one can understand that the big bang might not be the actual beginning of the universe. The same phenomenon can be viewed in the viewpoint of conformal cyclic cosmology. It stresses that the universe evolves like conformal cycles. By going through all possible evolution as backward time, the actual beginning of the universe seems to be meaningless in a local cycle. The question of actual beginning leads to the super universe hypothesis.

To understand the evolution at the beginning, a quantized curvature to be discussed. To prove the postulate I the obtained quantized curvature is reproduced from [21]

$$R = 6\left(\frac{\ddot{a}}{N^2\,a} + \frac{\dot{a}}{N^2\,a^2} + \frac{k}{a^2} - \frac{\dot{a}}{a}\frac{\dot{N}}{N^3}\right) \tag{3}$$

From the equation 3, the evolution of the universe, beyond the quantized big bang singularity can be predicted. As time-reversed evolution is confirmed one can understand the extension of the trajectory of evolution beyond the singularity.

Implementation of the conformal cyclic cosmological model with quantum cosmological analysis, the non-big bang beginning of the universe can be proved. The

equation 3 establishes the quantized curvature at the initial singularity. As the density varies with time the curvature also varies accordingly. In quantum cosmological scales, the time variable is taken as scalar field [7]. Such a paradigm stresses that the curvature evolution can be extended beyond the singularity and the big bang may not be the actual beginning as well. If the big bang may not be the actual initial point, then one can also predict that the future singularity also may not be the end of any universe. To predict the evolution beyond the initial singularity loop quantum cosmology and conformal cyclic cosmological model are introduced in the whole picture. The conformal analogy is added because it allows the universe to evolve cyclically. Since the universe might have evolved cyclically in time, then the current universe can be a continuation of the conformal extension of the previous universe. Hence one can prove that the universe might not begin from the big bang singularity. Also, the existence of Hawking points [16] provide evidence for the conformal evolution.

## 5.1 super universe hypothesis

The super universe hypothesis is proposed for postulate I. The laws of physics suggest the existence of a possible beginning and end of each successive aeons. But as a whole, the evolution of each successive aeons can be discussed as different lapses. For $N = x$ to $N = x'$ the transformation from an aeon to successive aeon takes place. In the same picture, the evolution of the parallel aeon also to be embedded to understand the entire evolution. The parallel evolution takes place at different time intervals. As the light pulses from an aeon can not reach the parallel aeon, one can not exactly predict the initial stages of the super universe without any known laws of physics. The problem lies in the initial value problem of the universe. In the super universe model one can estimate the lapse as $N = 0$, and let the calculations done on that perspective. But still the question remains what could be the possible states of the evolution beyond $N = 0$.

The beginning of an aeon at the lapse $N = 0$ is not a valid argument for this model. Every aeon can have begun at local time $t = 0$. But for the super universe hypothesis, the beginning, as well as the end, said to be meaningless. The evolution in the super universe hypothesis adds the parallel evolution of the aeons. Every aeon can have various local time. Hence they begin and end, independent of parallel aeons. The super universe model is portrayed in figure 1.

In the super universe hypothesis the time evolution can be written as

$$\tau = \int_{-\infty}^{-\infty} N \tag{4}$$

Here $\tau$ is the proper time and $N$ is the lapses. It is also written as

$$d\tau^2 = \sum_{N=-\infty}^{\infty} dN^2 + dX^2 \tag{5}$$

If the lapse intervals vary for various parallel universes, then the understanding of evolution will further be complicated. Perhaps some solutions from quantum gravity such as wormholes, white holes provide the answer for the evolution of

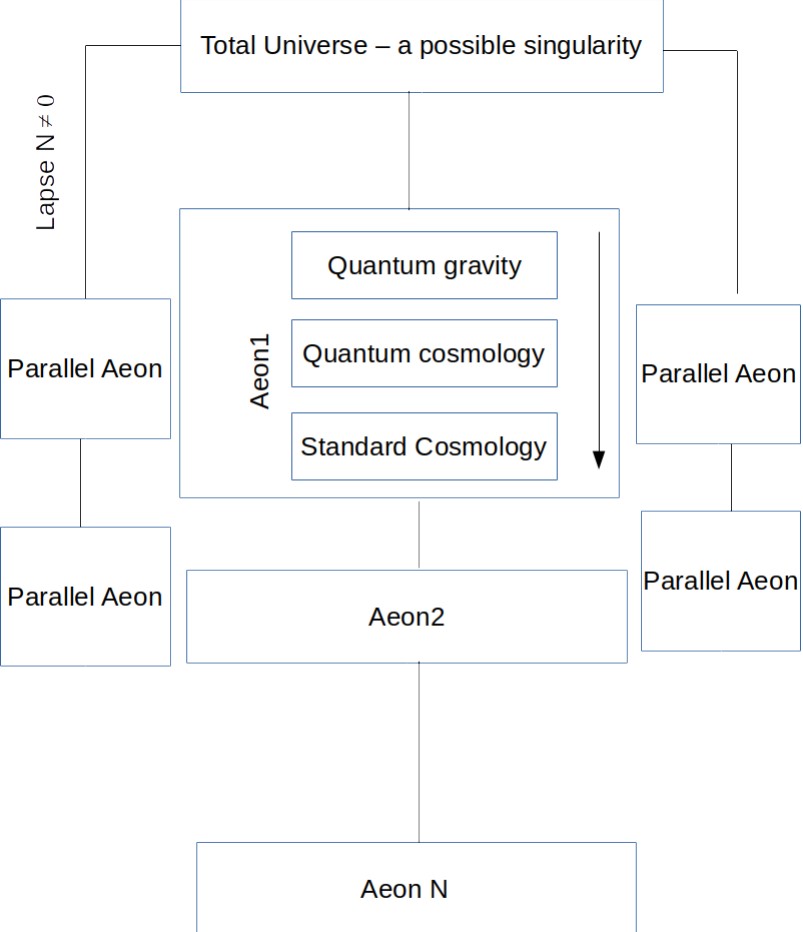

Figure 1: The super universe model. Paralell evolution of Aeons in various local time is shown.

parallel aeon. Because wormholes and white holes extend through the spacetime singularities. Hence it has been understood that it is meaningless to ask about beginning and end in the super universe model.

# 6  Proof for postulate II

As said earlier the expansion of the universe begins with the big bang, but one can inevitably ask what will be the end of the universe?.

The universe may have various possible ends. Among them, the leading candidates are big crunch and big rip. In the big crunch, the universe reaches again to a singular point like the big bang. To reach a big crunch the universe attains a critical density and its volume tends to shrink. Gradually the universe will reach the Planck sized big crunch singularity. In big crunch singularity, the matter density and pressure diverges as the scale factor vanishes.

During the evolution of the universe, the existence of big rip singularity is inevitable. At the big rip singularity in addition to the scale factor, the pressure and density parameters also diverge. The big rip singularity will be caused by the dominance of phantom energy (Which is a form of dark energy) in the future universe. Phantom energy has equation of state as $w < -1$. The phantom energy works against the gravitational pull and that let the universe expand exponentially. Time to rip off from the present scale of the universe is calculated as $(t - t_{br})$. The phantom energy will rip off the galactic clusters, galaxies, solar system and even the subatomic structures too. The various period to rip off is predicted in [10].

At the big rip, the universe will eventually come to the end of its evolution. As every constituent of the universe is ripped off completely, it can not extend its evolution beyond the big rip singularity. Hence the big rip is predicted as the theoretical end of the evolution of the universe. But the proposed postulate predicts that the big rip will not be the end of the universe. To prove this postulate, resolutions from loop quantum gravity and conformal cyclic cosmology are introduced. The loop quantum gravity deals with the big rip singularity as like other singularity resolutions [19]. But after the future big rip singularity is resolved, the evolution of the universe will be different evolution other than the classical big bang kind of resolution.

The critical loop quantum density [35] obtained from phantom energy will mimick the quantum gravity effects at the big rip singularity. The quantum gravitational effects induce the quantum cosmological scenario at that phase. But the evolution of the universe will be rephrased from a big rip. A unique way of expansion may occur beyond the big rip. To solve this crisis the conformal cyclic cosmology rescues the evolution with the alternative phenomenon. As the phantom density reaches the critical density the non-ending conformal aeon bypasses the big rip singularity. Hence a quantum bounce for the future universe is plausible. The scale factor at the time of the bounce is reproduced from [30].

$$a = \left(\frac{3}{4}\rho_c t^2\right)^{\frac{1}{3}} \left[1 + \tan^2\left[(\frac{3}{4}\rho_c t^2 + 1)^{\frac{1}{3}}\right] \frac{3V_0}{2M_P^2}(t - t_{br})^2\right] \tag{6}$$

By understanding the scale factor evolution from the equation 6, one can under-

stand the evolution of the quantum universe beyond the big rip singularity. Hence it is confirmed that the big rip and its evolution will be dominated by quantum gravitational effects. In addition to that, the universe beyond the big rip will have alternative possible quantum cosmological evolution. Beyond the big rip, it is also confirmed that the universe may evolve into higher dimensions as proven in [20]. This relationship also confirms the postulate II, and it is also suggested that the universe will not end its evolution by facing a future big rip. For both proofs of postulate II, the evolution is supported with the backend from conformal cyclic cosmology. Hence these two postulates eventually provide solid shreds of evidence for CCC. In the cyclic universe, the main issue is entropy. The entropy of the universe seems ever-growing. But for the conformal universe, there may be a small possible entropic reversal at the transition 2 surfaces. The possible entropic reversal is shown in [30].

The entropy of the super universe model increases ever. The core idea of the conformal cyclic cosmology avoids the reduction of entropy. Hence a minimal entropy of a cycle can not be less than that of maximal entropy of the previous aeon.

$$S_{C1}^{max} \geq S_{C2}^{max} \tag{7}$$

If these postulates combined, they prove and support the existence of CCC. As suggested earlier the universe may exit into higher dimensions to avoid the big rip. The classical and quantum gravitational solutions combined to produce such results. The FRW metric for such higher dimensions is found in [20].

The FRW equation is computed as

$$H^2 = \frac{8\pi G \rho(t)}{3} + \frac{\Lambda}{3} - \frac{2k}{a(t)^2} \tag{8}$$

The acceleration equation is computed as

$$\frac{\ddot{a}}{a} = -\frac{4\pi G \rho}{3}(\rho - 16) + \frac{\Lambda}{3} \tag{9}$$

Equation 8 represents the FRW equation for the conformal loop quantum universe. But the solution requires the implementation of loop quantum cosmology to represent the quantum cosmological solutions. Hence equation 8 is modified for $k = 0$ with negligible cosmological constant.

$$H^2 = \frac{8\pi G}{3} \rho \left(1 - \frac{\rho}{\rho_c}\right) \tag{10}$$

The equation 10 retrives the loop quantum bounce solutions and the accelleration equation is understood with fullfledged densiy domination. Solutions of the higher dimensional scale factor is explained in [20]. The scale factor for such non big rip aeon is also reprecented as refp2.3.1.

The equations 8 and 9 are also verified by the Higher-dimensional interpretation. These calculations are verified with Sagemanifolds [13], [8] mathematical package. The conformal evolution beyond the big rip is proposed and confirmed by postulate II. The phantom dominated big rip universe will conformally exit into future aeons without any perturbations.

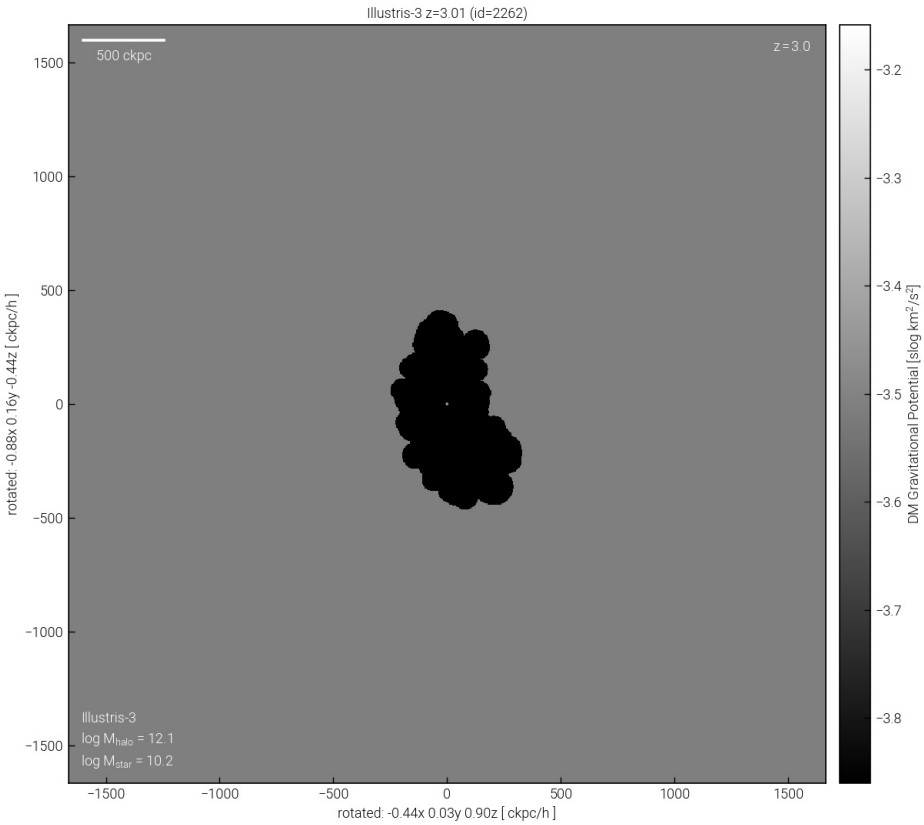

Figure 2: Dark matter gravitational potential interaction with $Z = 3.01$

# 7 Proof with datasets

To understand the big rip, few large scale data structures are required. Phantom energy is a form of dark energy which causes the universe to be ripped off in later times and on larger scales. To avoid the big rip the eternal universe to have non-interaction solutions between phantom energy and dark matter. The non-interacting solutions are discussed in [21]. With such non-interacting solutions ($r = 0$,$C = 0$) the scale factor approaches maximum and the universe avoids the big rip singularity. To analyse the behaviour of dark matter and dark energy in large scales, the higher scale dark matter gravitational interaction with high redshift is predicted. For such calculations the cosmological data are obtained from IllustrisTNG database [23] [28] [32] [22].

The role of dark matter and gravitational potential in a large scale universe with the redshift $Z = 3.01$ is plotted in figure 2.

The contour exits in between $-500$ to $+500$ and $-500$ to $+500$ in $X$ and $Y$ axis respectively. Among them, the maximal gravitational potential beyond $-3.8$ is understood from the graph. The graph shows that the elongation is greater than $1500\ CKPC$. Hence the interaction of the gravitational potential with dark energy might provide some glimpses for the existence of higher dimensions. Here it has been shown that the gravitational attraction of dark matter prevails instead of phantom interaction. This may confirm the results obtained from [21].

Using the IllustrisTNG data the gravitational interaction of dark matter with

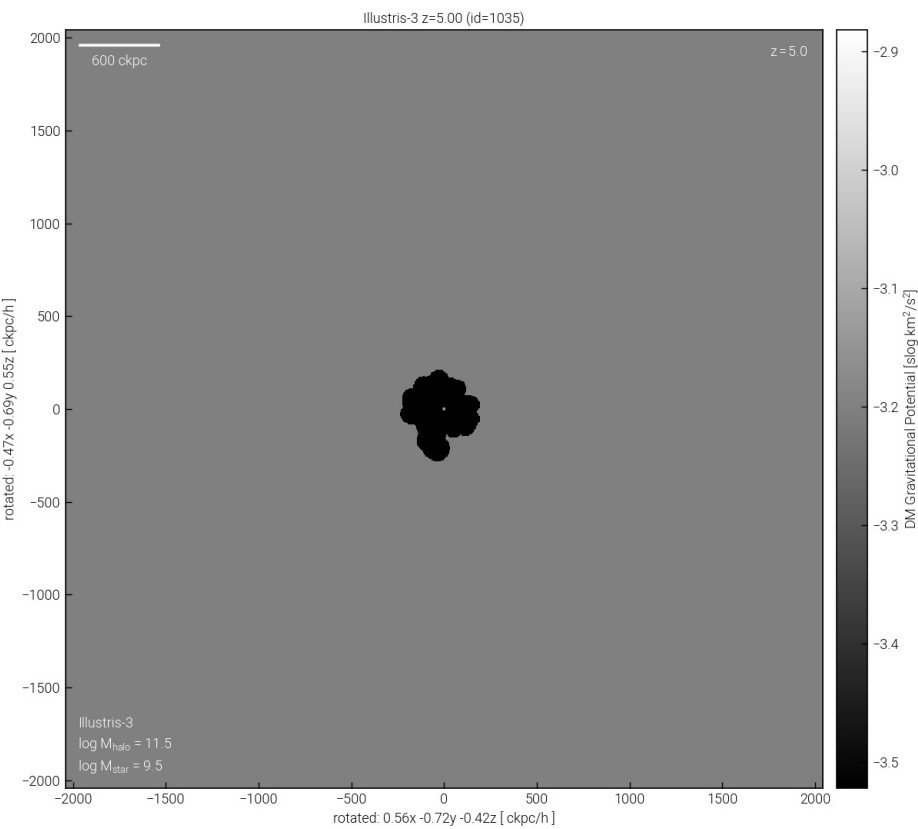

Figure 3: Dark matter gravitational potential interaction with $Z = 5.00$

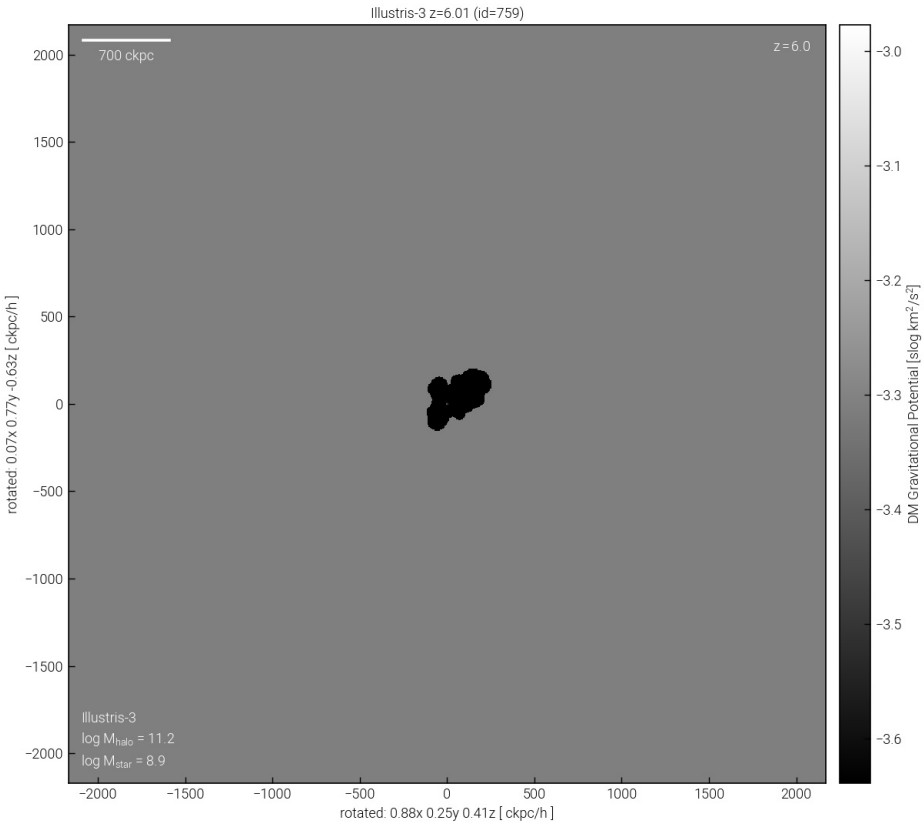

Figure 4: Dark matter gravitational potential interaction with $Z = 6.01$

further higher redshift ($Z = 5.0$) is plotted in figure 3. The plot shows the decrement in contour area compared to the image 2. From the plot, it has been observed that the contour area has changed around 700 $CKPC$. Beyond that level, the gravitational potential behaves normally.

Using the IllustrisTNG data the gravitational interaction of dark matter with further higher redshift ($Z = 6.01$) is plotted in figure 4. From the plot is has been observed that the central contour looks further lower in its area as the redshift increases. The shrinkage patterns can be analyzed on these three plots. It has been understood that the central contour area of the plot shrinks as the redshift increases. That observation means the clumps of dark matter are not perturbed by the expansion of the universe. If the dark matter is perturbed by the dark energy, then the contour will also have an increased area. By comparing the plots (2,3 and 4) one can find analysis, as the redshift increases the contour area of gravitational potential increases for the corresponding dark matter density. This shows the domination of dark matter over dark energy. It is also confirmed that the non-interaction between the dark energy and dark matter led the contour to shrink for the higher redshifts. Hence the increasing scale factor may avoid the future big rip and it may continue its evolution with the support of loop quantum cosmology. From the results provided above, one can confirm that the future big rip will not be the finite end of the expanding universe. Instead, its evolution will be continued even beyond the big rip. These datasets embedded here provides the possible scenario for the avoidance of big rip by introducing the dark matter concentrations at the

large scale universe.

# 8 Conclusion

The universe might have originated and evolved even without the existence of the big bang singularity. The known laws of physics will not allow us to extend our knowledge beyond the Plack regime. And the observations will not allow us to extend our wisdom beyond the event horizon of our known universe. Some theories like Conformal Cyclic Cosmology extend our observations through the evolutions of the aeons. One such attempt is proof of the existence of Hawking points. Such existence may open new windows on the observations of large scale astronomy. Even though new theories may require to understand the whole picture of the universe. The whole picture might consist of hidden dimensions, parallel universes, the reality of dark matter and dark energy, quantum gravity and yet other unknown theories of physics.

With the lapses in the understanding and observations on the evolution of the universe, the theories which we have on cosmological evolution may be incomplete. Hence it is premature to say that the big bang is the actual beginning of the universe. The proofs for this postulate is detailly discussed in various sections of the current work. The quantum effects preserve the evolution of the late time universe. As the complete rip off of the universe is avoided, the universe evolves even beyond such singularity. As a whole, the two postulates proposed here, are united towards a general platform namely the conformal cyclic cosmology. Such a model might provide a general understanding of the whole picture of the universe. From the analysis provided here, an evolution of the whole picture of the universe might be predicted.

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
