# Peer review of "Proposals on Conformal Cyclic Evolution of The Universe"

_SciPost Astronomy_

## Round 1 · Referee Report · Rathinam Chandramohan · 2021-12-16

Report

The author proposed two Cosmological postulates based on conformal cyclic cosmology. He has correlated loop quantum gravity and conformal cyclic cosmology in this work.

But For these postulates, proofs sketched by the authors is unsatisfactory.

The following questions arises which are not addressed.

1. Is there any possible consequences from string theory in this model?
2. In postulate 1, the authors suggested that the big bang may not be the actual beginning of the universe. But how does the time-reversal happen beyond t<0?
3. What is the role of inflation in these postulates?

The work " Proposals on Conformal Cyclic Evolution of The Universe " is rejected for publication in SciPost.

---

## Editorial Decision

editor-in-charge_assigned